# Changes in gene expression of cervical collagens, metalloproteinases, and tissue inhibitors of metalloproteinases after partial cervical excision-induced preterm labor in mice

Hyun Chul Jeong[1☯‡], Ho Yeon Kim[1☯], Hee Youn Kim[1], Eun-Jin Wang[1], Ki Hoon Ahn[1‡*], Min-Jeong Oh[1], Byung Min Choi[2], Hai-Joong Kim[1‡*]

1 Department of Obstetrics and Gynecology, Korea University College of Medicine, Seoul, Korea,
2 Department of Pediatrics, Ansan Hospital, Korea University College of Medicine, Ansan, Korea

☯ These authors contributed equally to this work.
‡ These authors also contributed equally to this work.
* haijkim@korea.ac.kr (HJK); akh1220@hanmail.net (KHA)

## Abstract

We investigated changes in gene expression of cervical collagens, matrix metalloproteinases (MMPs), and tissue inhibitors of metalloproteinases (TIMPs) during pre-gestational uterine cervical excision and/or inflammation-induced preterm labor in mice. Forty sexually mature female mice were uniformly divided into four groups: sham, cervical excision, lipopolysaccharide (LPS) injection, and cervical excision plus LPS injection. Partial cervical tissue excision was performed at five weeks of age before mating. LPS was injected into the lower right uterine horn near the cervix on gestational day 16. Mice were sacrificed immediately postpartum. Uterine cervices were collected and subjected to quantitative real-time PCR. Col4α1 and Col5α1 expression increased significantly in the cervical excision plus LPS injection group compared to the sham group ($p < 0.01$ and $p = 0.024$, respectively). MMP-14 expression levels increased in the cervical excision plus LPS injection group compared to the sham group ($p < 0.01$). TIMP-1 expression was not significantly decreased in this group. Increased expression levels of Col4α1, Col5α1, and MMP-14 were associated with cervical excision plus inflammation-induced preterm labor. Thus, pre-gestational cervical remodeling through specific collagen metabolism and MMP activation may involve the pathogenesis of spontaneous preterm labor.

## Introduction

The worldwide incidence of preterm birth is 11.1%, and the rates range from approximately 5–18% by country and region [1]. Preterm birth is one of the most common causes of neonatal morbidity and mortality [2]. The etiology of preterm birth is not yet fully understood. Infection, vascular disorders, decidual senescence, uterine overdistension, decline in progesterone

**Data Availability Statement:** All relevant data are within the manuscript and its Supporting Information files.

**Funding:** This research was supported by the Basic Science Research Program through the National Research Foundation of Korea, which is funded by the Ministry of Science and ICT40 (2014R1A1A1002300 and 2018R1D1A1B07). The funders had no role in study design, data collection and analysis, decision to publish, or preparation of the manuscript.

**Competing interests:** The authors have declared that no competing interests exist.

action, cervical disease, breakdown of maternal–fetal tolerance, and stress are the proposed mechanisms of preterm birth [3]. Intrauterine infection is the most studied causal factor for preterm birth. Moreover, partial excision of the uterine cervix, such as conization or loop electrosurgical excision procedure, is known to increase the risk of preterm birth [4]. However, there has been limited research on the association between pre-gestational cervical excision and change in the cervix during the pregnancy and postpartum period.

The uterine cervix is primarily composed of extracellular matrix (ECM) and smooth muscles. The cervix is characterized by the remodeling of ECM as pregnancy proceeds [5–7]. Matrix metalloproteinases (MMPs) are proteolytic enzymes that degrade ECM components, such as collagens, and their activity is regulated by tissue inhibitors of metalloproteinases (TIMPs) and Extracellular Matrix Metalloproteinase Inducer (EMMPRIN) [8]. MMPs are involved in cervical dilation and the pathology of preterm labor. TIMPs are endogenous specific inhibitors that regulate the proteolytic activity of MMPs in normal and pathologic processes [9, 10]. We previously demonstrated that uterine cervical excision plus inflammation by LPS resulted in a higher rate of preterm birth and changed the proximal cervical muscle-to-collagen ratio in mice [11]. Based on this finding, we proceeded to elucidate the changes in collagen components. We hypothesized that there would be a change in cervical collagen metabolism during the physiologic process of pre-injured cervix-associated preterm birth. The aim of this study is to investigate the change in gene expression of various collagens, MMPs, and TIMPs in the cervix of the mouse model of preterm birth caused by partial cervical excision.

## Materials and methods

### Study design

All experiments were performed following the guidelines for animal experiments and with approval from the Committee for the Care and Use of Laboratory Animals at our university (KUIACUC2015117). The method of animal experimentation is described in our previous study [11]. All surgery was performed under isoflurane inhalation and all efforts were made to minimize suffering. Sexually mature female C57BL/6 mice (n = 40) were randomly assigned into four groups with 10 mice in each group as follows: sham, partial cervical tissue excision, administration of LPS, and partial cervical tissue excision plus administration of LPS. The experimental room was maintained at a constant temperature (22–24°C) with a 12 h light/dark cycle. Food and water were supplied *ad libitum*. Before mating, five mice were allocated in one cage. Female and male mice were allowed to mate overnight and the presence of vaginal mucus plug on the next day confirmed mating. After mating, one pregnant mouse was allocated per cage for delivery care. The day when the mucus plug was found in the vagina was set as gestational day 1. In our preliminary experiments, the earliest optimal time to mate the animals for conception was at approximately 3 weeks after cervical tissue excision [11]. We found that even 3 weeks after cervical tissue excision, there was no difference in conception rate any further; therefore, mating was performed at 8 weeks of age. The conception rate through mating on the same day was 50% (20 out of 40 mice; 20 pregnant mice were randomly allocated into groups of partial cervical tissue excision and partial cervical tissue excision plus administration of LPS, with 10 mice in each group). LPS (100 μg from *Escherichia coli* 055:B5 [L6959, Sigma-Aldrich, St. Louis, MO, USA]) was used for the induction of inflammation. Body weight was measured preoperatively and during the postpartum period. Preterm birth was defined as deliveries within 24 h of LPS injection or before gestational day 18. Delivery was monitored continuously and recorded using a camera with night vision (EU-4124BI, Netinfo, Uiwang, Korea). The mice were euthanized immediately after completion of birth to evaluate the change in labor period as soon as possible. The time interval between delivery of the first and

the last pup was approximately 2 h. The uterine cervices were sampled for subsequent analyses.

## Cervical excision

Partial cervical tissue excision was performed at five weeks of age before mating. The mice were anesthetized with 2–4% isoflurane (Forane Solution, Choongwae, Seoul, South Korea) inhalation. For cervical tissue excision, the vaginal approach was performed with intravenous xylazine (Rompun®, Bayer, Toronto, Canada) as a pain control. The cervix was grasped with fine forceps, and a 1 mm depth excision was made using a scalpel. The same depth of external cervix was cut. Hemorrhage from the excision site was minimal, and the compression was sufficient for hemostasis. The length of the entire mouse cervix was approximately 2.5 mm, and approximately 40% of the cervix (1 out of 2.5 mm) was excised. Three weeks after partial cervical tissue excision, mating was performed when excision does not influence conception rate.

## Administration of LPS

LPS (100 $\mu g$ from Escherichia coli 055:B5 [L6959, Sigma-Aldrich, St. Louis, MO, USA]) was administered on gestational day 16. Anesthesia was administered as mentioned in the Cervical excision section. Minilaparotomy was performed, and the uterus was exteriorized. LPS was injected into the uterine cavity between the first and second gestational sacs of the right uterus [12]. In sham group, only anesthesia was administered, and no other procedure was performed.

## Real-time PCR analysis

Quantitative real-time PCR was performed to measure the relative mRNA expression levels of collagens, MMPs, and TIMPs. The cervix tissue was lysed with TRIzol solution (QIAzol lysis reagent, 79306, Qiagen, Valencia, CA, USA) and then homogenized with disposable homogenizer (Biomasher-II, 890864, Biomasher, Nippi, Tokyo, Japan). Total RNA was isolated by centrifugation ($14,000 \times g$ for 4 min at 4˚C). Amplification was performed on a thermocycler (C1000 Touch, Bio-Rad, Hercules, California, USA). RNA was reverse transcribed into cDNA using cDNA Synthesis Kit Hyperscript RT premix (GeneAll, Seoul, South Korea) according to the manufacturer's recommendations. Quantification of cDNA targets was performed using TOPreal qPCR 2X Premix (SYBR Green with low ROX, Enzynomics, Daejeon, South Korea). Real-time PCR was detected by the CFX96 Touch System (Bio-Rad, Hercules, California, USA). The primer sequences are listed in Table 1.

**Table 1. Primers for PCR analysis.**

| Genes | Forward primer sequence (5´→3´) | Reverse primer sequence (5''→3´) |
|---|---|---|
| GAPDH | TTG AGG TCA ATG AAG GGG TC | TCG TCC CGT AGA CAA AAT GG |
| Col1α1 | TAG GCC ATT GTG TAT GCA GC | ACA TGT TCA GCT TTG TGG ACC |
| Col2α1 | TGT CAT CGC AGA GGA CAT TC | CGG TCC TAC GGT GTC AGG |
| Col3α1 | TAG GAC TGA CCA AGG TGG CT | GGA ACC TGG TTT CTT CTC ACC |
| Col4α1 | CAC GCC ATG ACA GTC ACA TT | GTC TGG CTT CTG CTG CTC TT |
| Col5α1 | GTG GTT TTC GTT ACC CCT GA | GCT ACT CCT GTT CCT GCT GC |
| MMP-2 | CAA GTT CCC CGG CGA TGT C | TTC TGG TCA AGG TCA CCT GTC |
| MMP-7 | AGG AAG CTG GAG ATG TGA GC | TCT GCA TTT CCT TGA GGT TG |
| MMP-14 | CAG TAT GGC TAC CTA CCT CCA G | GCC TTG CCT GTC ACT TGT AAA |
| TIMP-1 | CTT GGT TCC CTG GCG TAC TC | ACC TGA TCC GTC CAC AAA CAG |

## Statistical analysis

The normal distribution was checked using the Shapiro-wilk test and the results are expressed as means ± standard deviations. Analysis of variance was used to compare the data between groups using Tukey's test and the Duncan method for multiple comparisons. Differences were regarded as statistically significant when p-values were less than 0.05. Data analysis was performed using SPSS 22.0 software (SPSS Inc., Chicago, IL, USA).

## Results

The mean gestational periods were 19.7 ± 1.2, 20.1 ± 0.5, 18.7 ± 1.5, and 17.6 ± 0.5 days in the sham(n = 10), partial cervical tissue excision(n = 9), administration of LPS(n = 5), and partial cervical tissue excision plus administration of LPS groups(n = 7), respectively. Moreover, as described in our previous study [11], the partial cervical tissue excision plus administration of LPS group showed a significantly shorter pregnancy period compared with the sham group. Gene expression levels of various collagens, MMPs, and TIMPs in cervix were examined according to groups.

The expression of collagens is described in Fig 1. Col1α1 and Col3α1 expression increased significantly in the administration of LPS group compared to the sham group. Although the expression levels of Col1α1 and Col3α1 tended to increase in the partial cervical tissue excision plus administration of LPS group, they were not statistically significant. Col4α1 expression increased significantly in the partial cervical tissue excision plus administration of LPS group compared to the sham and partial cervical tissue excision groups ($p < 0.01$ and $p = 0.047$, respectively). Expression of Col4α1 appeared to be higher in the administration of LPS group than in the sham group ($p = 0.023$). Col5α1 expression was higher in the partial cervical tissue excision plus administration of LPS group relative to the sham group ($p = 0.024$).

MMPs and TIMPs regarding Col4α1 and Col5α1 were investigated. MMP-7 expression was significantly higher in the administration of LPS group compared to the partial cervical tissue excision group, and MMP-14 expression level was significantly higher in the partial cervical tissue excision plus administration of LPS group relative to the sham and partial cervical tissue excision groups ($p < 0.01$ and $p = 0.015$, respectively; Fig 2A–2D). MMP-14 expression was significantly higher in the administration of LPS group compared to the sham group. There were no significant differences in the expression of MMP-2 and TIMP-1 among the experimental groups, but MMP-2 expression showed an increasing trend in the partial cervical

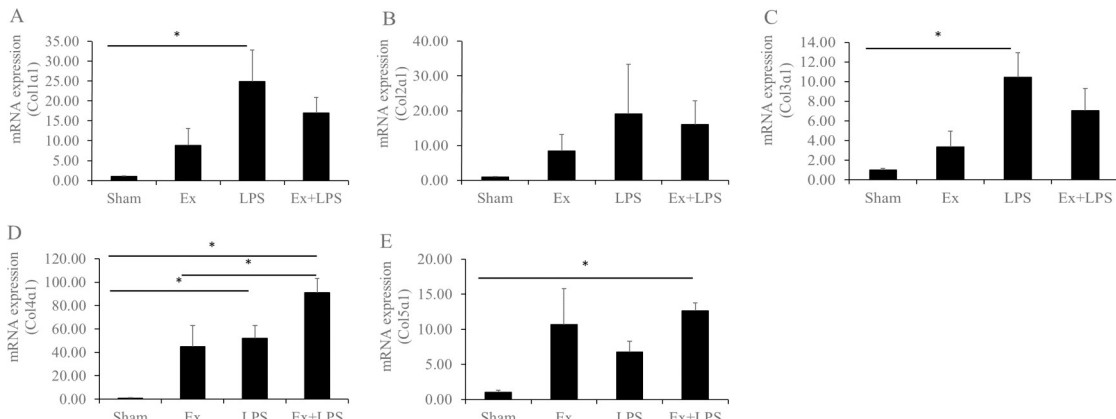

**Fig 1. mRNA expression levels of the collagens, metalloproteinases, and tissue inhibitors of metalloproteinases.** Data are presented as fold change. A) Col1α1, B) Col2α1, C) Col3α1, D) Col4α1, and E) Col5α1. *Statistically significant.

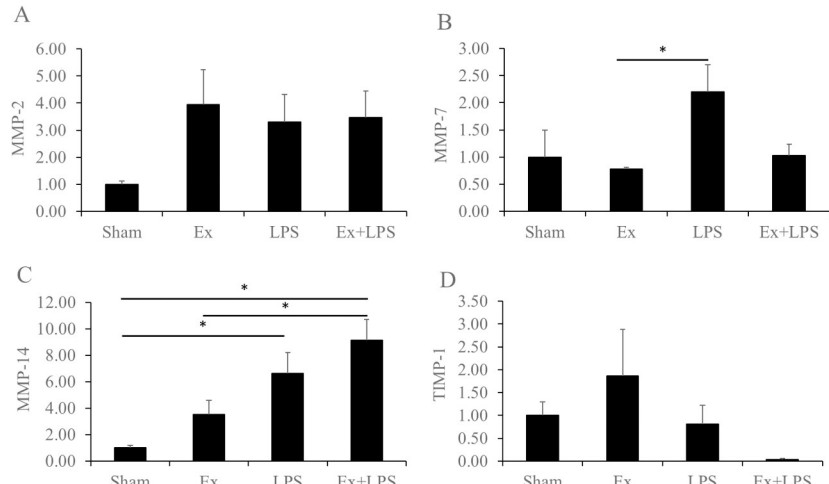

**Fig 2. mRNA expression levels of the matrix metalloproteinases and tissue inhibitors of metalloproteinases.** Data are presented as fold change. A) MMP-2, B) MMP-7, C) MMP-14, and D) TIMP-1. *Statistically significant.

tissue excision, administration of LPS, and partial cervical tissue excision plus administration of LPS groups compared to the sham group. In addition, TIMP-1 expression did not show significant differences but its expression in the partial cervical tissue excision plus administration of LPS group was lower compared to the other groups.

## Discussion

In our previous study, it was demonstrated that the cervical muscle-to-collagen ratio significantly changed in excision-induced preterm labor [11]. Since there has been minimal research to determine collagen changes in the remaining cervical tissue after pre-gestational partial cervical excision, levels of various collagens, MMPs, and TIMPs related to preterm birth were examined in this pilot study. Interestingly, this study showed that Col4α1 and Col5α1 expression were significantly increased in the cervix following partial cervical tissue excision and administration of LPS-mediated inflammatory stimuli, leading to cervical remodeling and subsequent preterm birth (Fig 3). Another notable observation was that MMP-14 expression also increased significantly in this group.

The cervix is composed of ECM, mainly collagens, which maintain cervical tissue strength biochemically and mechanically. The mechanical strength of cervical ECM is dependent on the type and degree of collagen cross-linking in its collagen network [13]. However not much research on various collagen types and their expression has been done in cervical remodeling and preterm or term birth. We for the first time explored that there were certain changes in collagen expression when the cervix was infected and traumatized even more with infection. Our results can be linked to the difference in distribution depending on the types of collagen, and findings of previous studies also support this idea [14, 15]. It is a key determinant in the assembly of cervical tissue-specific collagen matrices, such as those formed by Col1 and Col3. Immunohistochemistry analysis of pregnant mouse cervices showed that Col4 is located in the basement membrane and distributed in a linear fashion, delineating individual muscle fibers, but Col1 and Col3 collagens are distributed primarily around smooth muscle fiber bundles [15]. Col5 is located in the vicinity of the basement membrane [14]. Although the basal levels of these collagens in pre-injured cervix were higher than those in sham cervix, the differences were not significant and manipulation of cervix might result change in collagen expression.

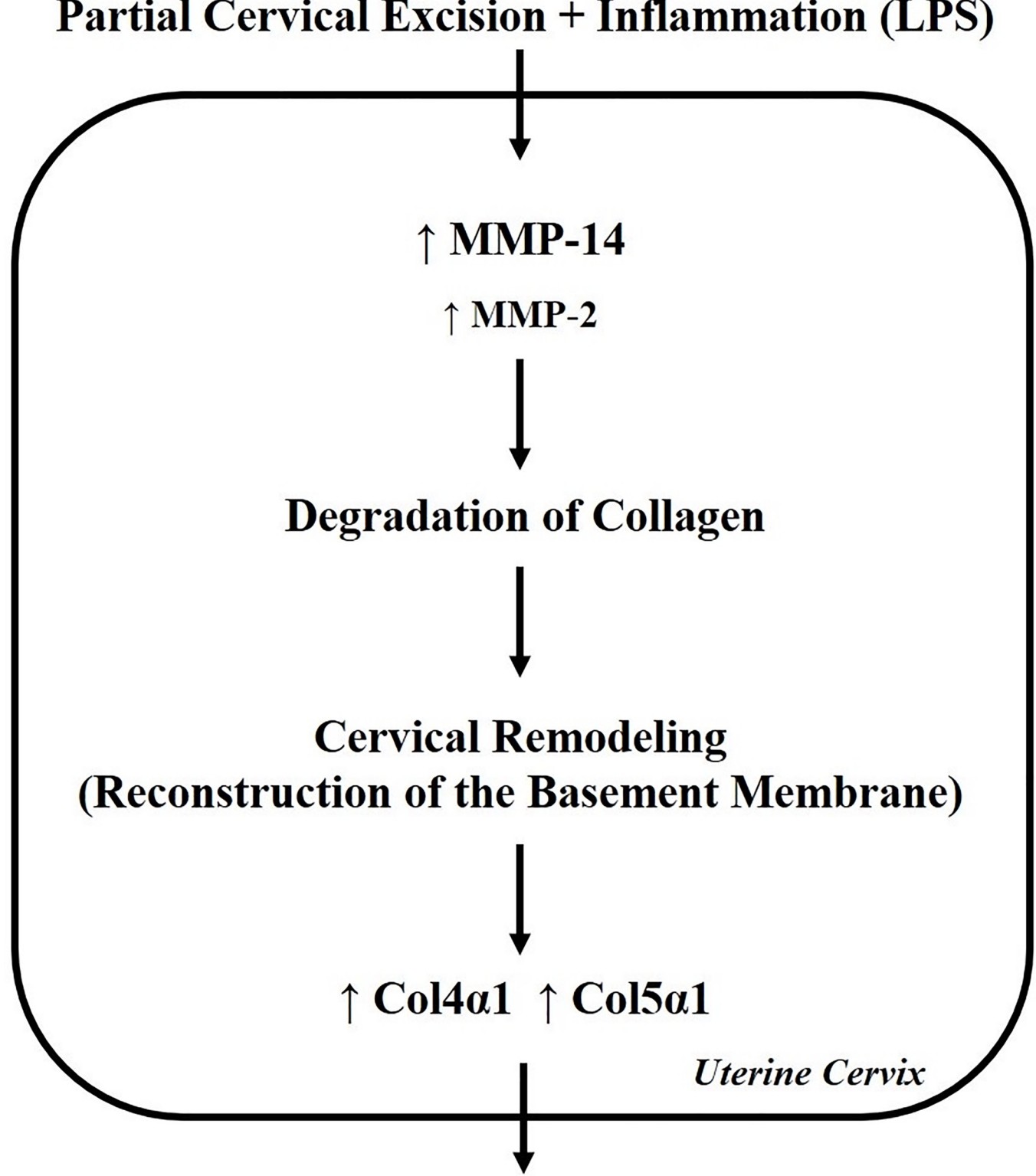

**Fig 3. Theoretical model for the role of collagens and MMPs in cervical partial excision plus LPS administration.** ECM remodeling may result from higher MMP activities followed by degradation of collagen. This may increase expression of collagen4α1 and collagen5α1 in cervix. Thus, pre-gestational cervical remodeling through specific collagen metabolism and MMP activation may involve the pathogenesis of spontaneous preterm labor.

The injury of cervix could induce the increase in the level of messenger RNA of collagens to build up more collagens for tissue remodeling of cervix.

In the present study, it was observed that Col4α1 and Col5α1 play a specific role in cervical remodeling from partial excision plus LPS-mediated inflammation-induced preterm labor. It is postulated that Col4 and Col5 are needed in the reconstruction of the basement membrane after inflammation-induced preterm labor following cervical tissue injury. Another possible explanation for this observation is that cervical excision may exert an additive effect on LPS-mediated cervical change or initiate another pathway. However caution should be taken on the roles of these Col4 and Col5 because there was no preterm birth in partial excision group even though these collagen expression was elevated up to same level as LPS-treated group.

In general, MMPs are not expressed in healthy tissues but rather in inflamed or damaged tissues [16]. In this case, white blood cells use both MMPs and TIMPs to penetrate various tissues of pregnant women, releasing cytokines and chemokines to create a sterile inflammatory environment and commencing labor [17]. Therefore, early increased expression of MMPs is associated with premature labor, and MMPs may play a direct role in the initiation of labor [18]. In a previous study, the proportion of preterm labor was reduced in mice injected with MMP inhibitors [19]. Although understanding which MMPs or TIMPs are responsible for cervical remodeling and membrane rupture remains unclear, we conducted this study to elucidate the function of MMP2, MMP-7, MMP-14 and TIMP-1 which are closely related to Col4 and Col5 utilizing our cervical excision model [20–22]. We observed higher basal levels of MMP-2, MMP-14 and TIMP-1 in partial excision, LPS-induced and partial excision plus LPS induced group compared to sham and this phenomenon might be attributed to cervical manipulation itself irrespective of preterm birth. But MMP-7 expression was quite different from others that MMP-7 is one of matrilysins while MMP-2 is one of gelatinases and MMP-14 is one of membrane-type MMPs therefore its function and targets might be different [23, 24]. MMP7 is recruited to the plasma membrane of epithelium inducing membrane-associated growth factors processing for epithelial repair and proliferation. Because cervical excision reduces the volume of cervical epithelium, the tissue reaction of cervical epithelium might be decreased.

Our results identified significantly increased expression of MMP-14 and non-significantly decreased expression of TIMP-1 in the cervical excision plus LPS-mediated inflammation-induced model. MMP-14 is one of the membranous type MMPs primarily located on the extracellular side of the cell membrane (cell surface) found in uninucleate trophoblasts of bovine and uterine interstitial cells in rats [24, 25]. MMP-14 is the key player in trophoblast invasion and main activator of MMP-2, which degrades type IV collagen and increases during parturition [24, 26, 27]. This function may explain our finding that the role of TIMP-1 is compromised in order to activate MMP-14. In our experiment, unlike for MMP-14 expression, there was no significant difference between groups in MMP-2 expression. Therefore, the increased expression levels of MMP-14 in this study suggest that MMP-14 may act as a *de novo* initial trigger and an additive factor for MMP2 pathways, leading to cervical remodeling and subsequent preterm birth. It is likely that preterm delivery shares the same mechanism as term delivery but occurs through a different pathway.

Regarding TIMP-1 expression, there have been some controversies with respect to preterm labor. Some studies have reported, consistent with our results, that TIMP-1 level increases in the cervicovaginal fluid during term pregnancy labor and in preterm birth with cervical

insufficiency [28]. However, other studies have shown that the levels of TIMP-1 in serum and amniotic fluid decrease during preterm labor or at uterine contraction [29, 30]. This discrepancy may be due to the difference in the localization of enzyme inhibitors, cervical tissue, uterine myometrium, serum, amniotic fluid, or cervicovaginal fluid. For example, TIMP-1 also increases in the amniotic fluid with intrauterine infection and is observed during labor and LPS-induced premature labor in the mouse uterus [31, 32]. Various MMP and TIMP polymorphisms are thought to be associated with preterm birth [33].

Based on the results of the present study, it is postulated that pre-pregnancy cervical excision exerts an additive effect on inflammation-induced preterm birth, indicating that tissue excision acts as a trigger to amplify the reaction leading to inflammation-associated preterm birth. Cervical excision reduces the volume of the cervix and may subsequently weaken its resistance to deformation against gravity and the pressure exerted by the enlarging gravid uterus. Furthermore, the shorter the cervix is during pregnancy, the more prone it is to microbial invasion, which can lead to preterm birth due to infection [34]. As the uterine cervix shortens, the total number of immune cells in the cervical tissue decreases, and the cervix becomes more vulnerable to infection. As a result, the pregnant uterus may become prone to infection and inflammatory reactions, thereby accelerating preterm labor. Endogenous and exogenous cervical defects, including excision plus inflammation, may play a crucial role in the underlying pathologic process of collagen remodeling in preterm labor. But human data have indicated no increased preterm birth after cervical excision such as conization with knife or laser previously [35–38]. These evidences explain no preterm birth after partial excision of cervix although the expression of Col4 and Col5 was reached to LPS induced group in this study. This suggests that partial excision without inflammation might be different from natural cervical ripening process.

The novelty of the present study is the determination of collagen, MMPs, and TIMPs expression in the cervix in a direct cervical excision plus LPS-mediated inflammation-associated preterm birth animal model. However, there are limitations of this study. For one, it is primarily descriptive and lacks explanation of how the observed changes occurred. As mentioned earlier, little is known about the changes in the cervix ECM after excision. For this reason, we examined only the expression of mRNA by real time PCR as the first step therefore immunoblot study including zymography is lacking. In addition, small number of mice were used especially in LPS induced group therefore inconsistent results might make confusions regarding expression of collagens and MMPs. MMP7 expression was lower in excision plus LPS induced group compared to LPS induced group and MMP-2 expression was elevated while TIMP-1 expression was elevated at the same time in the LPS induced group. More number of study subjects are needed. Further studies on the properties of proteins or changes in the activity of enzymes before and after cervical excision with periodic sampling would contribute to broaden the knowledge of the mechanisms leading to preterm labor.

Elevated mRNA expression of Col4$\alpha$1, Col5$\alpha$1 and MMP-14 was demonstrated in cervix after partial excision plus administration of LPS. Collagens are main component of cervix and their crosslink and relationship to other factors influencing cervical remodeling/ripening have been widely studied. However studies on the expression of mRNA of different types collagens and its relationship to MMPs are limited in the field of preterm birth. Their function might be a clue to solve the problem with cervical change and subsequent preterm birth. Much remains to be determined to fully elucidate cervical roles in labor and how cervical defects plus inflammation lead to preterm birth. Investigations to identify specific collagens and its related factors influencing cervical changes may provide clinically relevant targets for early detection and prevention of preterm birth.

## Supporting information

**S1 Data.**
(XLSX)

## Author Contributions

**Conceptualization:** Hyun Chul Jeong, Ho Yeon Kim, Ki Hoon Ahn, Hai-Joong Kim.

**Data curation:** Hee Youn Kim.

**Formal analysis:** Hee Youn Kim.

**Funding acquisition:** Ki Hoon Ahn.

**Investigation:** Hyun Chul Jeong, Eun-Jin Wang.

**Methodology:** Ho Yeon Kim, Hee Youn Kim, Eun-Jin Wang.

**Project administration:** Ki Hoon Ahn.

**Resources:** Byung Min Choi.

**Supervision:** Eun-Jin Wang, Min-Jeong Oh, Byung Min Choi, Hai-Joong Kim.

**Validation:** Ho Yeon Kim, Min-Jeong Oh, Byung Min Choi, Hai-Joong Kim.

**Writing – original draft:** Hyun Chul Jeong, Ho Yeon Kim, Ki Hoon Ahn.

**Writing – review & editing:** Min-Jeong Oh, Hai-Joong Kim.

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
