## [Decision Letter · Decision Letter 0]

19 Jan 2021

PONE-D-20-36937

Changes in gene expression of cervical collagens, metalloproteinases, and tissue inhibitors of metalloproteinases after partial cervical excision-induced preterm labor in mice

PLOS ONE

Dear Dr. Ahn,

Thank you for submitting your manuscript to PLOS ONE. After careful consideration, we feel that it has merit but does not fully meet PLOS ONE’s publication criteria as it currently stands. Therefore, we invite you to submit a revised version of the manuscript that addresses the points raised during the review process.

We look forward to receiving your revised manuscript.

Kind regards,

Carlos Zaragoza, Ph.D.

Academic Editor

PLOS ONE

Journal Requirements:

"This research was supported by the Basic Science Research Program through the

39 National Research Foundation of Korea, which is funded by the Ministry of Science and ICT

40 (2014R1A1A1002300 and 2018R1D1A1B07)."

Reviewers' comments:

Reviewer's Responses to Questions

**Comments to the Author**

1. Is the manuscript technically sound, and do the data support the conclusions?

Reviewer #1: Partly

Reviewer #2: Yes

2. Has the statistical analysis been performed appropriately and rigorously? 

Reviewer #1: Yes

Reviewer #2: Yes

3. Have the authors made all data underlying the findings in their manuscript fully available?

Reviewer #1: Yes

Reviewer #2: Yes

4. Is the manuscript presented in an intelligible fashion and written in standard English?

Reviewer #1: Yes

Reviewer #2: Yes

5. Review Comments to the Author

Reviewer #1: To test the changes in the levels of cervical collagens MMPs and TIMPs, it would be convenient for the authors to perform immunoblot with the available samples.

In the line 121, when the authors mention their previous studies in the methods, they should include a reference if they are published.

When the authors explain the cervical excision, it would be convenient included if any kind of analgesia has been administered to the animals after the surgery.

In the results, in line 175 authors should include the SD in the gestational periods.

On the y-axis of the figures, authors should include the name of the collagenase, MMP or TIMs studied instead of fold.

In figure 2 authors should check the letters off the graphs

Reviewer #2: In the present manuscript the authors show the changes in the expression of cervical collagens, metalloproteinases and tissue inhibitors of metalloproteinases that are produced in a model of preterm labor induced by a partial cervical excision in mice. Thus, the authors justify that these components and proteins participate in the remodeling after a spontaneous premature birth after inflation added to a cervical excision.

However, the study lacks of novelty. The comments are shown below.

Introduction

Minor comments

- The authors should show that it is not only TIMPs that are regulating MMPS activity. There are other proteins that can participate in this signaling pathway, such as CD147, also known as EMMPRIN (Extracellular Matrix Metalloproteinase INducer). Line 94.

- Paragraph staring from line 86 to 90 is necessary to remove it. Experimentation and research on humans are totally forbidden and does not need to be discussed.

Methods

- What is the concentration of LPS given to the mice?

- Why was the administration of LPS done on the 16th day gestational?

Results

- Figure 2 shows graphically significant differences in the expression levels of MMP2 and TIMP-1 that the authors do not comment.

- Why does the LPS group have a low n compared to the other groups?

- Figure 2, the graphics are incorrectly listed

- Due to the fact that there are a wide number of Metalloproteinases and TIMPs involved in remodeling processes in response to an injury, the authors should screen for these proteins (MMP-9, TIMP-2, MMP-13...)

- To complete the study, the authors must show the protein expression of the MMPs through Western Blot and must also show the activity of these proteins using a Zymography

- In relation to line 299 to 301, the authors should use for basic techniques like Hematoxylin/Eosin or Masson's Trichrome stain to show a change in the cervix ECM before/after excision

6. PLOS authors have the option to publish the peer review history of their article (what does this mean?). If published, this will include your full peer review and any attached files.

Reviewer #1: No

Reviewer #2: No

---

## [Author Response · Author response to Decision Letter 0]

24 Feb 2021

Reviewer's Responses to Questions

Comments to the Author

1. Is the manuscript technically sound, and do the data support the conclusions?

Reviewer #1: Partly

Reviewer #2: Yes

2. Has the statistical analysis been performed appropriately and rigorously?

Reviewer #1: Yes

Reviewer #2: Yes

3. Have the authors made all data underlying the findings in their manuscript fully available?

Reviewer #1: Yes

Reviewer #2: Yes

4. Is the manuscript presented in an intelligible fashion and written in standard English?

Reviewer #1: Yes

Reviewer #2: Yes

5. Review Comments to the Author

Response to Reviewer

Reviewer #1: To test the changes in the levels of cervical collagens MMPs and TIMPs, it would be convenient for the authors to perform immunoblot with the available samples.

We truly aware of your critical advice. Immunoblot would make our data more understandable however we tried to quantify gene expression of cervical collagens, MMPs and TIMPs using our cervical excision mouse model. Now our team don’t have available samples because this project already ended. Spite of the absence of protein expression data, this study might be meaningful as a pilot study. This limitation has been inserted as a limitation in Discussion section. Thank you for your understanding.

In the line 121, when the authors mention their previous studies in the methods, they should include a reference if they are published.

We added the reference of our prior experiment.

11. Ahn KH, Jeong HC, Kim HY, Kang D, Hong SC, Cho GJ et al. Relationship between partial uterine cervical tissue excision and preterm birth: an experimental animal study. Am J Perinatol. 2017;34:1072–1077.

When the authors explain the cervical excision, it would be convenient included if any kind of analgesia has been administered to the animals after the surgery.

For the cervical excision surgery, xylazine (Rompun®, Bayer, Toronto, Canada) is administered intravenously immediately before surgery. This information has been added in Method section. (Line 102-103)

In the results, in line 175 authors should include the SD in the gestational periods.

As per comment, we have added the SD value in the gestational periods.

On the y-axis of the figures, authors should include the name of the collagenase, MMP or TIMs studied instead of fold.

We included the name of the collagenase, MMP or TIMPs on the y-axis of the figures as suggested instead of fold. 

In figure 2 authors should check the letters off the graphs

We corrected as suggested in Figure 2.

Reviewer #2: In the present manuscript the authors show the changes in the expression of cervical collagens, metalloproteinases and tissue inhibitors of metalloproteinases that are produced in a model of preterm labor induced by a partial cervical excision in mice. Thus, the authors justify that these components and proteins participate in the remodeling after a spontaneous premature birth after inflation added to a cervical excision.

However, the study lacks of novelty. The comments are shown below.

Introduction

Minor comments

- The authors should show that it is not only TIMPs that are regulating MMPS activity. There are other proteins that can participate in this signaling pathway, such as CD147, also known as EMMPRIN (Extracellular Matrix Metalloproteinase INducer). Line 94.

As per comment, we added EMMPRIN as a regulator of MMPs with a reference. 

8. Li K, Nowak RA. The role of basigin in reproduction. Reproduction. 2019 Sep 1:REP-19-0268.R1.

- Paragraph staring from line 86 to 90 is necessary to remove it. Experimentation and research on humans are totally forbidden and does not need to be discussed.\\

As per comment, we deleted sentences in line 86 to 90. 

Methods

- What is the concentration of LPS given to the mice?

We gave 100 μg of LPS in each mice. We added more detailed explanation for volume of LPS concentration in Material and Methods section. 

- Why was the administration of LPS done on the 16th day gestational?

Preterm birth is defined before gestational day 18 in mouse therefore LPS induction is usually performed on gestational day 15-17 to induce preterm birth. In addition, this protocol using LPS induced preterm birth is usually performed on gestational day 15-17 in previous studies. Following reference shows similar protocol. 

The Local and Systemic Immune Response to Intrauterine LPS in the Prepartum Mouse. Edey LF, O'Dea KP, Herbert BR, Hua R, Waddington SN, MacIntyre DA, Bennett PR, Takata M, Johnson MR. Biol Reprod. 2016 Dec;95(6):125. doi: 10.1095/biolreprod.116.143289. Epub 2016 Oct 19.

Results

- Figure 2 shows graphically significant differences in the expression levels of MMP2 and TIMP-1 that the authors do not comment.

We made comments on MMP2 and TIMP-1 in results line 167-173 as follows. 

There were no significant differences in the expression of MMP-2 and TIMP-1 among the experimental groups, but MMP-2 expression showed an increasing trend in the partial cervical tissue excision, administration of LPS, and partial cervical tissue excision plus administration of LPS groups compared to the sham group. In addition, TIMP-1 expression did not show significant differences although its expression in the partial cervical tissue excision plus administration of LPS group was lower compared to the other groups.

- Why does the LPS group have a low n compared to the other groups?

Thank you for your comment. To minimize the conditions affecting the experiment, mouse experiments were conducted at once. The mice were randomly assigned to one of four groups and distributed 10 mice per group. Mice was bred in the eighth week, and mice that were not pregnant for more than eight weeks were sacrificed. The LPS groups had fewer mice pregnant at week 8 compared to other groups. 

- Figure 2, the graphics are incorrectly listed

We corrected Figure 2.

- Due to the fact that there are a wide number of Metalloproteinases and TIMPs involved in remodeling processes in response to an injury, the authors should screen for these proteins (MMP-9, TIMP-2, MMP-13...)

We appreciate suggestion on screening for MMP-9 and other MMPs and TIMPs which have been widely studied in abortion, preeclampsia, and preterm birth. We conducted this study to elucidate the function of MMP-2, MMP-7, MMP-14 and TIMP-1 which are closely related to Col4 and Col5 utilizing our cervical excision model. We focused on the Col4 and Col5 and related enzymes in this study.

- To complete the study, the authors must show the protein expression of the MMPs through Western Blot and must also show the activity of these proteins using a Zymography

Thank you for your comment. Although the current study is focusing on the gene quantification of collagens, MMPs and TIMPs of partial cervical excision plus LPS, authors agree with the reviewer’s viewpoint. We addressed your points as a limitation in the Discussion line 284-285.

- In relation to line 299 to 301, the authors should use for basic techniques like Hematoxylin/Eosin or Masson's Trichrome stain to show a change in the cervix ECM before/after excision

Thank you for your comment. In our study, the cervical excision and LPS injection is one treatment option, which is compared to sham group. Therefore, change in the collagen and its enzymes after the treatment (cervical excision and LPS injection as a whole) is important. We have a future plan for the study on ECM change before vs. periodic follow-up after excision. Your advice is commented as a limitation in the main text.

6. PLOS authors have the option to publish the peer review history of their article (what does this mean?). If published, this will include your full peer review and any attached files.

Do you want your identity to be public for this peer review? For information about this choice, including consent withdrawal, please see our Privacy Policy.

Reviewer #1: No

Reviewer #2: No

---

## [Decision Letter · Decision Letter 1]

23 Mar 2021

PONE-D-20-36937R1

Changes in gene expression of cervical collagens, metalloproteinases, and tissue inhibitors of metalloproteinases after partial cervical excision-induced preterm labor in mice

PLOS ONE

Dear Dr. Ahn,

Thank you for submitting your manuscript to PLOS ONE. After careful consideration, we feel that it has merit but does not fully meet PLOS ONE’s publication criteria as it currently stands. Therefore, we invite you to submit a revised version of the manuscript that addresses the points raised during the review process.

We look forward to receiving your revised manuscript.

Kind regards,

Carlos Zaragoza, Ph.D.

Academic Editor

PLOS ONE

Journal Requirements:

Reviewers' comments:

Reviewer's Responses to Questions

**Comments to the Author**

1. If the authors have adequately addressed your comments raised in a previous round of review and you feel that this manuscript is now acceptable for publication, you may indicate that here to bypass the “Comments to the Author” section, enter your conflict of interest statement in the “Confidential to Editor” section, and submit your "Accept" recommendation.

Reviewer #1: All comments have been addressed

Reviewer #2: All comments have been addressed

2. Is the manuscript technically sound, and do the data support the conclusions?

Reviewer #1: Yes

Reviewer #2: Yes

3. Has the statistical analysis been performed appropriately and rigorously? 

Reviewer #1: Yes

Reviewer #2: Yes

4. Have the authors made all data underlying the findings in their manuscript fully available?

Reviewer #1: Yes

Reviewer #2: Yes

5. Is the manuscript presented in an intelligible fashion and written in standard English?

Reviewer #1: Yes

Reviewer #2: Yes

6. Review Comments to the Author

Reviewer #1: Authors should check in line 117 of the new manuscript if the reference to the previous study is number 10 or 11 "The method of animal experimentation is described in our previous study".

Reviewer #2: All comments have been addressed  correctly. The authors have made the manuscript better and improved the quality of the work.

7. PLOS authors have the option to publish the peer review history of their article (what does this mean?). If published, this will include your full peer review and any attached files.

Reviewer #1: No

Reviewer #2: No

---

## [Author Response · Author response to Decision Letter 1]

24 Mar 2021

Response to editor

Thank you for your positive decision.

Response to reviewers

Reviewer #1: Authors should check in line 117 of the new manuscript if the reference to the previous study is number 10 or 11 "The method of animal experimentation is described in our previous study".

As per your comment, we have corrected the number of the reference. (11 instead of 10)

Reviewer #2: All comments have been addressed correctly. The authors have made the manuscript better and improved the quality of the work.

Thank you for your positive answer.

---

## [Editor Report · Decision Letter 2]

31 Mar 2021

Changes in gene expression of cervical collagens, metalloproteinases, and tissue inhibitors of metalloproteinases after partial cervical excision-induced preterm labor in mice

PONE-D-20-36937R2

Dear Dr. Ahn,

We’re pleased to inform you that your manuscript has been judged scientifically suitable for publication and will be formally accepted for publication once it meets all outstanding technical requirements.

Kind regards,

Carlos Zaragoza, Ph.D.

Academic Editor

PLOS ONE
---

## [Editor Report · Acceptance letter]

5 Apr 2021

PONE-D-20-36937R2 

Changes in gene expression of cervical collagens, metalloproteinases, and tissue inhibitors of metalloproteinases after partial cervical excision-induced preterm labor in mice 

Dear Dr. Ahn:

I'm pleased to inform you that your manuscript has been deemed suitable for publication in PLOS ONE. Congratulations! Your manuscript is now with our production department. 

Kind regards, 

on behalf of

Dr Carlos Zaragoza 

Academic Editor

PLOS ONE